# Thin Hydrogenated Amorphous Silicon Carbide Layers with Embedded Ge Nanocrystals

**DOI:** 10.3390/nano15030176

**Published:** 2025-01-23

**Authors:** Zdeněk Remeš, Jiří Stuchlík, Jaroslav Kupčík, Oleg Babčenko

**Affiliations:** FZU—Institute of Physics of the Czech Academy of Sciences, Na Slovance 1999/2, 182 00 Prague, Czech Republic; stuj@fzu.cz (J.S.); kupcik@fzu.cz (J.K.); babcenko@fzu.cz (O.B.)

**Keywords:** a-SiC:H, Ge NPs, PECVD, TEM, PDS, PL

## Abstract

The in situ combination of plasma-enhanced chemical vapor deposition (PECVD) and vacuum evaporation in the same vacuum chamber allowed us to integrate germanium nanocrystals (Ge NCs) into hydrogenated amorphous silicon carbide (a-SiC:H) thin films deposited from monomethyl silane diluted with hydrogen. Transmission electron microscopy (TEM) and energy-dispersive X-ray (EDX) spectroscopy were used for the microscopic characterization, while photothermal deflection spectroscopy (PDS) and near-infrared photoluminescence spectroscopy (NIR PL) were for optical characterization. The presence of Ge NCs embedded in the amorphous a-Si:C:H thin films was confirmed by TEM and EDX. The embedded Ge NCs increased optical absorption in the NIR spectral region. The quenching of a-SiC:H NIR PL due to the presence of Ge indicates that the diffusion length of free charge carriers in a-SiC:H is in the range of a few tens of nm, an order of magnitude less than in a-Si:H. The optical properties of a-SiC:H films were degraded after vacuum annealing at 550 °C.

## 1. Introduction

Thin films with embedded nanoparticles exhibit an enhanced spectral response in the near-infrared region [1]. Germanium nanocrystals (Ge NCs) embedded in thin layers improve light absorption in hydrogenated amorphous silicon (a-Si:H) solar cells [2]. Ge has a smaller band gap than Si (0.7 eV with respect to 1.1 eV) and a bigger excitonic Bohr radius (about 24 nm with respect to 5 nm) [3]. However, the formation of Ge NCs requires a relatively high temperature above 400 °C, whereas the optimal deposition temperature of a-Si:H is around 250 °C [4]. Therefore, we suggested replacing a-Si:H with amorphous sub-stoichiometric silicon carbide thin films with low carbon content prepared from a mixture of H_2_, SiH_4_, and CH_4_ by plasma-enhanced chemical vapor deposition at a temperature of about 400 °C [5]. Moreover, the optical band gap of a-SiC:H can be modified from 1.8 to 2.4 eV by varying the carbon content and making it suitable for applications requiring wide-band-gap materials such as passivation layers in photovoltaic applications to protect the underlying materials and improve the overall efficiency of solar cells [6,7].

The a-SiC:H thin films have a wide range of applications due to their unique optical and electrical properties such as a high photoconductivity, wide band gap, and good thermal stability [8]. SiC/Si was used in heterojunction diodes with high breakdown voltage [9], thin-film transistors [10], high-gain photodetectors [11], and various microelectronic applications including coatings and thin films for thermal barriers [12,13]. a-SiC:H also serves as a low-loss dielectric material in superconducting devices, such as integrated superconducting spectrometers and superconducting qubits [14], and various harsh environment applications [15]. The a-SiC:H is often used in multi-junction solar cells as a window layer where the addition of carbon increases the band gap [16] to improve efficiency by enhancing light absorption and reducing recombination losses [17]. The a-SiC:H thin films are utilized in advanced photonic applications and an integrated photonic platform, such as in the fabrication of ring resonators offering high-quality factors and low propagation losses [18].

In our previous works, we presented p–i–n structures based on a-SiC:H thin layers prepared by plasma-enhanced chemical vapor deposition (PECVD) from a mixture of silane and methane with embedded Ge NPs deposited ex situ by molecular-beam epitaxy [19]. Later, we demonstrated the plasma-enhanced chemical vapor deposition (PECVD) in combination with in situ vacuum evaporation (VE) and plasma treatment (PT) as a convenient technology for the deposition of amorphous thin films with integrated NPs [20,21]. We showed that the integration of Ge NPs can modify the electronic structure of a-SiC:H thin films in such a way that the electroluminescence (EL) intensity is strongly enhanced and the intensity of EL correlates with the current density [22].

This study aims to develop a novel synthesis method for a-SiC:H films embedded with Ge NPs and to investigate their structural and optical properties. The integration of Ge NPs into a-SiC:H films could potentially enhance their optical and electronic properties, making them suitable for advanced optoelectronic applications. While previous studies have explored the synthesis of a-SiC:H films and Ge NPs separately, there is limited research on the in situ synthesis of these materials in a single chamber, which could offer better control over the material properties. This study introduces a novel synthesis method that allows for the in situ formation of a-SiC:H films and Ge NPs, resulting in a unique layered structure with potentially superior properties.

## 2. Materials and Methods

### 2.1. PECVD, Vacuum Evaporation, and Thermal Annealing

The a-SiC:H thin films were deposited using PECVD at about 400 °C on fused silica substrates from monomethyl silane (CH_3_SiH_3_, Linde Gas, Praha, Czechia) diluted by hydrogen in the flow ratio of 2 to 100 sccm [23]. The stainless steel vacuum deposition chamber is shown in Figure 1 as a cylinder in capacitive configuration equipped with two electrodes (size: 62 × 62 mm^2^, distance: 32 mm, frequency: 13.56 MHz, and RF power: 18 W). The grounded sample holder (12) serves as a top electrode. The bottom electrode (10) was movable to allow in situ evaporation. The limit pressure after 12 h of heating and pumping reached a level of less than 2 × 10^−5^ Pa. The gas pressure during PECVD was 30 Pa. To incorporate Ge into a-SiC:H, the PECVD process was interrupted, the chamber evacuated and about 4 nm of Ge was deposited in situ by evaporation at a low pressure of 2 × 10^−5^ Pa. The thickness of Ge evaporated from a Tungsten boat was controlled by a Standard Quartz Crystal Sensor (MCVAC Manufacturing Co. Inc, East Syracuse, NY, USA). After Ge deposition, the PECVD process was completed to obtain Ge embedded in the a-SiC:H layer. To create Ge NCs in the sample #2-Ge, the a-SiC:H surface was saturated prior to the Ge evaporation with carbon using methane in hydrogen plasma (flow rate ratio: 100:2). Samples #1, #2, and #3 were prepared using the same process as #1-Ge, #2-Ge, and #3-Ge but covered with a shield (10) during Ge evaporation. A set of two substrates was always used for the deposition of a-SiC:H, one for the thin film with and the second without evaporating Ge. The samples for the TEM analysis were deposited onto standard (commercial) Cu TEM grids covered by carbon foil. Table 1 shows the list of samples.

The samples were annealed in a horizontal furnace (Clasic CZ, Ltd., Revnice, Czech Republic) with a corundum tube as a cylindric chamber. The furnace chamber was evacuated to the pressure 6 × 10^−2^ Pa and then heated to 550 °C (ramp: 3 °C/min), which was kept for 12 h. After annealing, samples were naturally cooled to 50 °C with constant chamber evacuation and removed to the ambient air.

### 2.2. Transmission Electron Microscopy (TEM)

The high-resolution TEM analysis was performed on a Tecnai TF20Xtwin HRTEM (Thermo Fisher Scientific, Waltham, MA, USA), equipped with a field-emission gun (FEG) cathode, an X-TWIN lens, a high-brightness field-emission electron gun operated at 200 kV, a high-angle annual detector (HAADF) usable in the scanning TEM (STEM) imaging mode, an energy-dispersive X-ray analyzer for an elemental analysis and mapping or profiling, a 4-Megapixel CCD camera for imaging in the convectional mode, a high-angle annular dark field detector working in the scanning mode, and an EDAX energy-dispersive X-ray spectrometer (EDX) system working over the range of 0–40 keV with a 20 eV per channel dispersion for an elemental composition analysis. Micrographs were processed by Digital Micrograph 3.20.1314.0 software (Gatan, Pleasanton, CA, USA). Samples for the TEM analysis were prepared under the same conditions as the samples deposited on fused silica (described above) but prepared directly on standard (commercial) carbon-coated Cu TEM grids.

### 2.3. Photothermal Deflection Spectroscopy (PDS)

The transmittance, reflectance, and absorptance spectra were measured simultaneously in the 300–1400 nm spectral range by a photothermal deflection spectroscopy (PDS) setup with a 150 W Xe lamp, and monochromator (SpectraPro-150, Acton Research Corp., Acton, MA, USA) equipped with two gratings: a UV holographic (1200/mm) and a ruled (600/mm) grating blazed at 500 nm and with slits at 1/1 mm [24]. The spectral resolution was 5 nm with the UV grating and 10 nm with the ruled grating. Samples were immersed during PDS into liquid (Florinert FC72) to measure the relative temperature of the illuminated sample independently for selected photon energies using the deflection of a probe laser beam. The spectra were spectrally calibrated by measuring PDS of a black carbon sample.

### 2.4. Photoluminescence Spectroscopy

The setup for NIR PL spectroscopy used in this work has evolved from our previous NIR PL setup [25]; see Figure 2. The blue LED (10 mW M470F2, Thorlabs, Newton, NJ, USA) was triggered by a pulse generator (Model 3390, Keithley Instruments, Inc., Cleveland, OH, USA) and collimated by lens L1 (Thorlabs, Newton, NJ, USA). To increase efficiency of collecting the PL signal from smooth thin layers, the sample was illuminated perpendicularly via the lens L2 and the same lens collected the PL signal and an interchangeable long-pass dichroic mirror (5 mm × 36 mm, 650 nm cut-on, Thorlabs DMLP650R with DFM1T3 Kinematic 30 mm Cage Cube Insert for Ø25 mm Fluorescence Filters, Thorlabs, Newton, NJ, USA) was placed between lenses L2 and L3, reflecting wavelengths below 650 nm and transmitting above 650 nm. Plano-convex lens L3 (L4) (LA5817, Thorlabs, Newton, NJ, USA) then focused (collected) light on (from) the slit of the monochromator. The excitation and emission were spectrally purified by the OD6 band pass (BP) and long pass (LP) optical filters (FBH470-10 and FELH0550, Thorlabs, Newton, NJ, USA). Emitted PL light was spectrally analyzed using the monochromator (f/4.2 aperture, 200 mm focal length, 600g/mm concave holographic grating at 400–1600 nm, slits at 1/1mm, spectral resolution at 8 nm, Horiba H20IR, Jobin Yvon SA, France) and focused by lens L4 on interchangeable Si and InGaAs photodiodes with built-in amplifiers (models 2151 and 2152, Newport Corp., Andover, MA, USA), connected to a lock-in amplifier (Model SR830, Stanford Research Systems, Sunnyvale, CA, USA) referenced to LED frequency (37 Hz). The spectral response was calibrated by a calibrated halogen lamp followed by an additional correction [26].

## 3. Results

### 3.1. TEM

The bright field TEM image in Figure 3a shows darker amorphous islands with sizes in the range of several nm embedded in the amorphous sample #1-Ge. The SAED (selected area electron diffraction) pattern (DP) in Figure 3b obtained from an area approx. 600 nm in diameter shows broad diffusive circles, typical for amorphous materials. Due to its amorphous character, it is not possible to distinguish crystal nuclei of Ge in the amorphous Ge phase from crystal nuclei present in the amorphous silicon carbide phase. On the other hand, the bright field TEM image and corresponding diffraction pattern in Figure 3c,d clearly show Ge NC inclusions in the sample #2-Ge with clearly visible lattice fringes embedded in the amorphous phase. DP can be assigned to crystalline Ge NCs with fcc cubic structure.

EDX analyses were obtained during the TEM analysis in the STEM mode. The presence of dissolved Ge homogeneously dispersed was confirmed in the sample #1-Ge. From these results, we conclude that Ge NCs failed to integrate into the a-SiC:H layer. On the other hand, the EDX profile in Figure 4a confirmed the higher occurrence of Ge in the oval inclusions (Ge NCs) in the sample #2-Ge (darker in BF TEM, lighter in the STEM image), while between them, it dropped to zero; see Figure 4c. High Ge concentration (Figure 4d) agrees well with the high HAADF intensity (Figure 4b), confirming that the oval particles are indeed composed of Ge whereas the Si-K line intensity was higher between spots and at the Ge NCs, it was lower but did not drop to zero. O and C showed no significant trends.

### 3.2. Optical Spectroscopy

Figure 5 shows the optical absorptance with and without Ge, before and after vacuum annealing at 550 °C. The as-grown a-SiC:H thin layers are highly transparent in NIR. Ge increased NIR absorptance in all samples. Ge NCs in the sample #2-Ge increased NIR absorption strongly even below 1.1 eV. After vacuum annealing at 550 °C, the optical absorption of a-SiC:H has increased significantly in the whole spectral range, indicating the deterioration of the a-SiC:H layer.

The optical absorption coefficient was evaluated from the optical spectra of a thicker sample, #3. First, the index of refraction was evaluated from the reflectance spectrum using the parametric Lorenc oscillator model and then the optical absorption coefficient was evaluated at each energy independently from optical absorptance spectra measured by PDS using the commercial thin-film software FilmWizard^®^ (https://sci-soft.com/product/film-wizard/, accessed on 1 January 2020) [27]. A Tauc gap of 2.15 eV before vacuum annealing was evaluated in the spectral region 2.4–3.0 eV corresponding to the optical absorption coefficient values from 10,000 to 70,000 cm^−1^ [28]; see Figure 6.

As-grown a-SiC:H thin layers exhibited a PL band in the NIR spectral range well below the optical absorption edge, 2.15 eV; see Figure 7. In all samples, PL was degraded by the presence of Ge in the a-SiC:H layer. The PL completely diminished due to the presence of Ge NCs in the sample #2-Ge and it was reduced to half in the thicker sample, #3-Ge. After vacuum annealing at 550 °C, the NIR PL has completely diminished in all samples. We attribute this effect to H diffusion out of the a-SiC:H layer [29]. Since hydrogen plays an important role in defect passivation, the effusion of H at elevated temperatures leads to a significant increase in localized defect states in the energy gap and the loss of the optical absorption edge [30].

## 4. Discussion

The optical absorption edge (Tauc gap) of the as-deposited intrinsic a-SiC:H layer was in the orange part of the light spectrum (energy: 2.15 eV, wavelength: 577 nm). Since the NIR PL band with the maximum at 1.3 eV (950 nm) was observed in very thin a-SiC:H film #2, we attribute this PL band to surface states. On the other hand, the NIR PL with the maximum at about 1.5 eV (830 nm) was attributable to the localized defect states in the bulk of thicker sample #3. In both cases, NIR PL decreased in samples with embedded Ge. Thus, we conclude that the free carrier diffusion length in a-SiC:H is in the range of a few tens of nm. This estimation follows the observation that PL diminished due to the presence of Ge-induced non-radiative recombination in the very thin sample #2-Ge (thickness of a-SiC:H sub-layer: 4 nm) whereas it was partly preserved in thicker sample #3-Ge (thickness of a-SiC:H sub-layer: 45 nm). If the diffusion length was larger, no NIR PL would be observed in the sample #3-Ge. The free carrier diffusion length is a critical parameter for performance in photovoltaic and photonic devices [31]. The diffusion length typically depends on deposition conditions, and the presence of defects or impurities. For example, hydrogen incorporated into the film during deposition or post-deposition annealing passivates dangling bonds, thereby increasing the diffusion length [29]. When exposed to light, additional defects are created in amorphous materials, known as the Staebler–Wronski effect, which reduces the diffusion length over time [32]. Previous studies have shown that the diffusion length in a-Si:H (a closely related material) is typically in the range of a few hundred nanometers [33]. For a-SiC:H, the values are expected to be lower due to the incorporation of carbon. Our estimate of diffusion length provides an alternative to methods such as photoconductivity transients [34].

## 5. Conclusions

It was not possible to achieve the post-deposition crystallization of Ge dissolved in a-SiC:H film by high-temperature annealing at 550 °C. We explain the a-SiC:H degradation at elevated temperature by the effusion of H, leading to a significant increase in optical absorptance and the loss of NIR photoluminescence. In addition, we have shown that the Ge NCs can be grown in situ at about 400 °C after preventing the dissolution of evaporated Ge in a-SiC:H by a thin carbon layer deposited on the a-SiC:H surface prior to Ge evaporation. The presence of Ge NCs embedded in the amorphous a-SiC:H layer was confirmed by TEM and EDX spectroscopy. The incorporation of Ge NCs in a-SiC:H thin layers extended the absorption spectrum into the near-infrared region, which is beneficial for capturing more NIR light and increasing photocurrent sensitivity. However, the decrease in intrinsic NIR PL showed that Ge acts as a non-radiative recombination center. In this paper, we proposed an estimation of the diffusion length in a-SiC:H thin layers based on the detection of photoluminescence quenching via embedded Ge. By comparing the NIR PL of a-SiC:H layers with different thicknesses, we concluded that the free carrier diffusion length in as-grown a-SiC:H was estimated to be in the range of a few tens of nm, about one order of magnitude lower than the diffusion length of a-Si:H.

## Figures and Tables

**Figure 1 nanomaterials-15-00176-f001:**
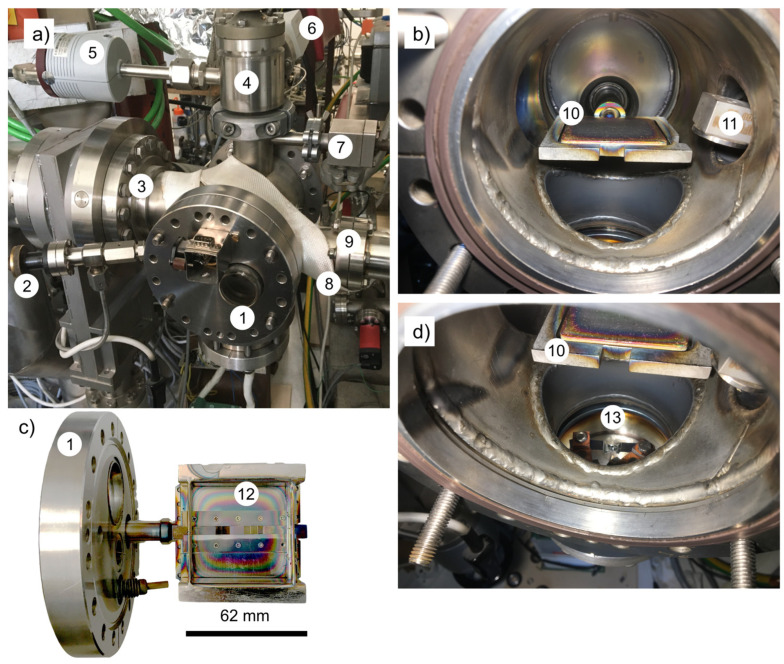
(**a**) The overall view of the PECVD chamber with the built-in evaporator, (**b**) detail of the interior, (**c**) front door with the sample holder, and (**d**) detail of the evaporator. Parts: A front flange of the chamber (1), a gas inlet to the chamber (2), a throttle valve for adjusting the pressure in the chamber (3), a shut-off valve between the chamber and the absolute pressure manometer (4), a diaphragm Baratron manometer for measuring the absolute pressure in the chamber (5), a manometer for measuring the vacuum level before deposition (6), a chamber filling valve after cooling the deposited samples after deposition (7), a heating belt for heating the chamber for total degassing (8), a flange for a quartz probe for measuring the thickness of the evaporated layer (9), a shutter and a lower electrode driven by high frequency for the excitation of the discharge (10), a quartz probe for measuring the evaporated layer (11), an earthed upper electrode and a heated sample holder (12), and a heated boat for Ge evaporation (13).

**Figure 2 nanomaterials-15-00176-f002:**
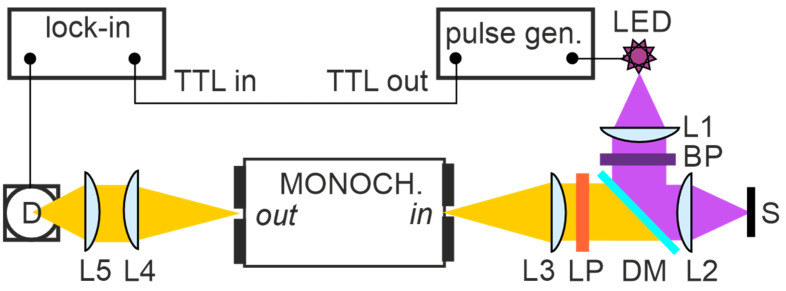
NIR PL setup with monochromator and LED excitation powered by pulse generator; plano-convex lenses L1, L3, and L4; aspheric lenses L2 and L5; dichroic mirror (DM); sample (S); band pass (BP) and long pass (LP) optical filters; detector (D); and lock-in amplifier referenced to pulse generator frequency by TTL. Focal lengths are 25 mm (L2 and L5), 50 mm (L1), and 100 mm (L3 and L4). Diameter of all lenses was 25 mm.

**Figure 3 nanomaterials-15-00176-f003:**
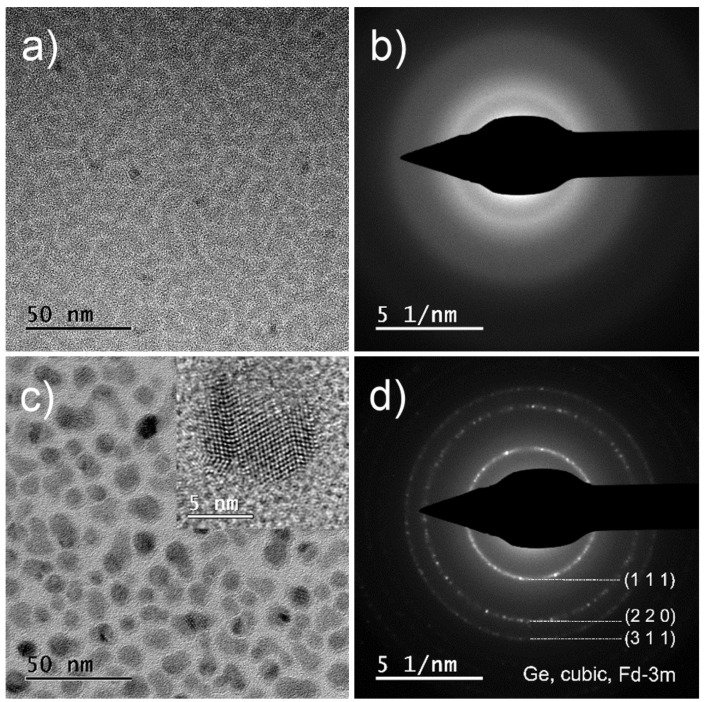
(**a**) The bright field TEM and (**b**) corresponding diffraction pattern of an a-SiC:H layer with dissolved Ge (the sample #1-Ge). (**c**) The bright field TEM of an a-SiC:H layer with embedded Ge NCs (the sample #2-Ge) with an enlarged HR-TEM picture of a single NC in the corner and (**d**) diffraction pattern of the sample #2-Ge with interpretation.

**Figure 4 nanomaterials-15-00176-f004:**
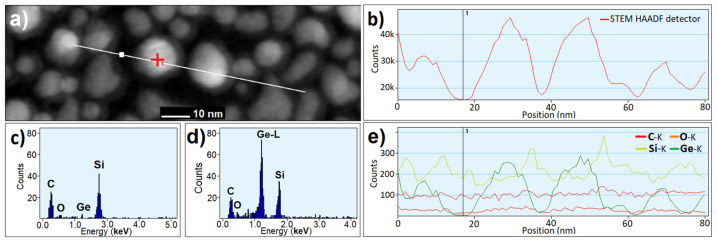
EDX elemental profile of sample #2-Ge obtained at line of 80 nm with step of 1 nm—(**a**): STEM image with profile line and points of depicted point spectra marked; (**b**) HAADF intensity measured simultaneously with EDX spectrum profile; (**c**) point EDX spectrum obtained at point marked with white point; (**d**): point EDX spectrum obtained at point marked with red cross; (**e**): occurrence of elements indicated by intensities of char. X-lines along profile.

**Figure 5 nanomaterials-15-00176-f005:**
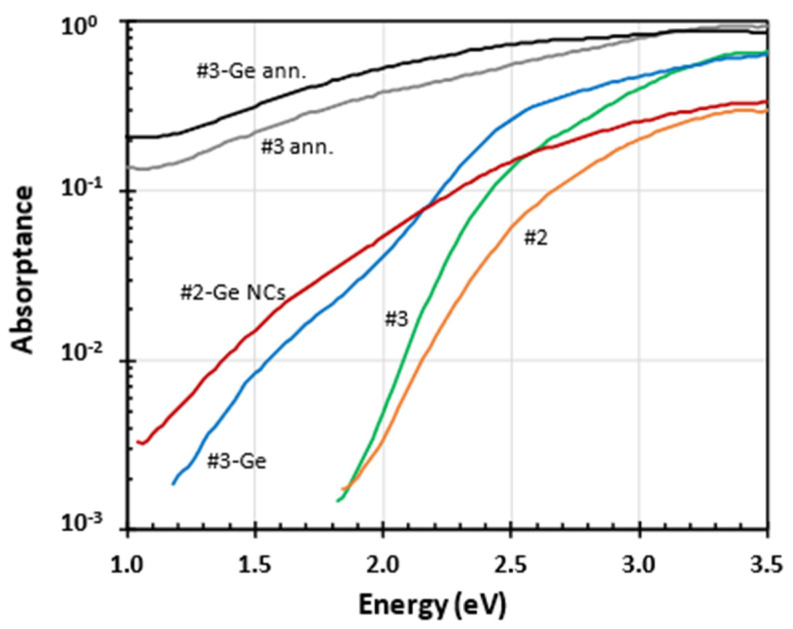
Optical absorptance spectra of a-SiC:H layers with and without Ge, before and after vacuum annealing (ann.) at 550 °C.

**Figure 6 nanomaterials-15-00176-f006:**
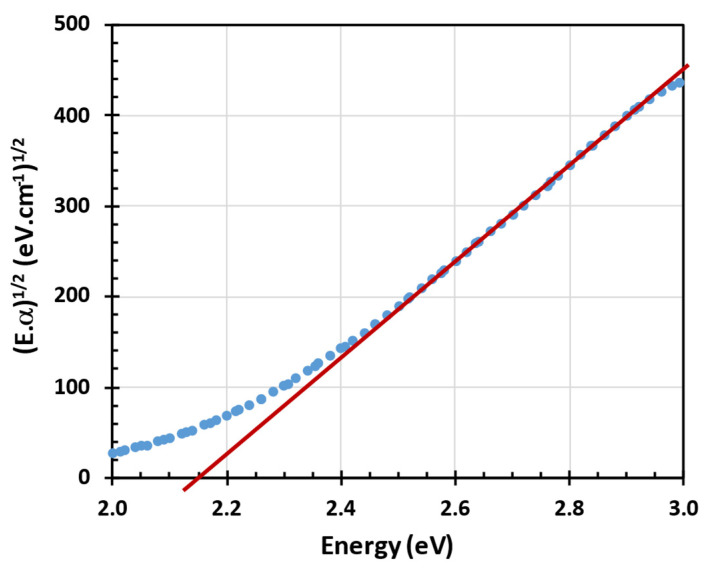
Optical absorption edge (Tauc gap) evaluation of 90 nm thick a-SiC:H layer without Ge (sample #3) before vacuum annealing.

**Figure 7 nanomaterials-15-00176-f007:**
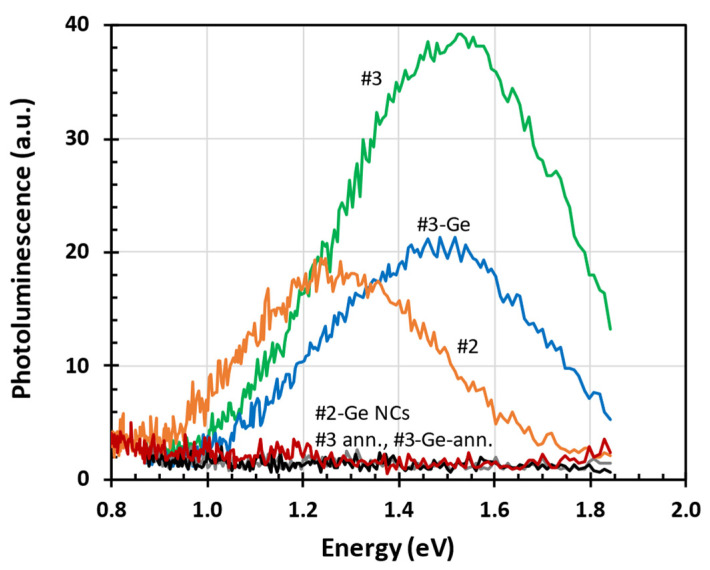
NIR PL spectra of a-SiC:H layers with and without Ge, before and after vacuum annealing (ann.) at 550 °C.

**Table 1 nanomaterials-15-00176-t001:** List of samples composed of thin amorphous a-SiC:H sub-layers (A), embedded Ge (G), and carbon (C) layers. The total thickness *d* and thickness of A, G, and C sub-layers are shown.

Sample	d (nm)	A (nm)	G (nm)	C (nm)	Structure	Ge Form
#1	8	4	0	0	AA	none
#2	16	4	0	0	AA	none
#3	90	45	0	0	AA	none
#1-Ge	12	4	4	0	AGA	dissolved
#2-Ge	21	8	4	1	ACGA	NCs
#3-Ge	94	45	4	0	AGA	dissolved

## Data Availability

The data are available in a public ASEP repository of the Czech Academy of Sciences (https://doi.org/10.57680/asep.0601980), accessed on 30 November 2024.

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
