# Peer review of "Thin Hydrogenated Amorphous Silicon Carbide Layers with Embedded Ge Nanocrystals"

_nanomaterials, 2025, doi:10.3390/nano15030176_

Round 1

Reviewer 1 Report

Comments and Suggestions for Authors

The manuscript titled "Thin hydrogenated amorphous silicon carbide layers with embedded Ge nanocrystals" by Z Remes et al describes the synthesis and characterization of hydrogenated amorphous SiC (a-SiC:H) films embedded with Ge nanoparticles (Ge NPs). The a-SiC:H films and Ge NPs are synthesized using different methods but in situ, in the same chamber, creating a layered structure. Structural analysis, including TEM, SAED, and EDX, is used to confirm the formation of Ge NPs, while optical properties are characterized using absorption spectroscopy and photoluminescence spectroscopy.

The introduction is not clearly written; it is difficult to understand the specific objectives and motivations behind the research. It is unclear whether similar systems have been studied before and what knowledge gaps the authors are addressing. Is this a new synthesis method, a material with improved properties, or a novel system with measurable advantages?

The composition of amorphous silicon the authors are interested in is not clear: what does x < 0,1 mean in the formula a-Si1-xCx:H? Does it imply 0.1? The authors then refer to the material as simply a-SiC:H, making comparison of the results with literature data problematic. What is the actual composition of the film used in the study?

The sentence "It is challenging to measure low values of diffusion length in amorphous thin layers." appears abruptly at the end of the introduction and requires further elaboration.

Further, the rationale behind the choice of film thicknesses and the resulting optical properties is not clear. Why do the thinnest film (#1-Ge) and the thickest film (#2-Ge) contain dissolved Ge NPs, while sample #3 does not? What is the mechanism? Can the dissolution of Ge NPs be controlled by thickness, or what causes different sample outcomes? Additionally, the optical properties of Sample #1 are not shown later in the manuscript.

The samples are not clearly identified in the manuscript. Table 1 presents only a few samples; it should be expanded to include all samples produced in this work for easier reference.

Figure 2, depicting the sample structure, does not indicate that a carbon layer was deposited before Ge deposition. Was another C deposition performed afterward before the next layer of a-SiC:H? How does this C layer affect the overall composition of the film and subsequently its optical properties?

Page 3, line 90: Which samples are being referred to as "selected samples"?

Figure 7 and the corresponding text: The authors use both "absorption" and "absorptance" when referring to the same effect. The terminology should be consistent.

Page 7: What is the purpose of annealing the film at 550 °C, and how does the conclusion about the effusion of H apply to the samples with Ge nanoparticles?

Page 7, line 212: The authors should elaborate on how they determined the specific range of 10-30 nm for the free carrier diffusion length.

Author Response

REVIEWER I

The manuscript titled "Thin hydrogenated amorphous silicon carbide layers with embedded Ge nanocrystals" by Z Remes et al describes the synthesis and characterization of hydrogenated amorphous SiC (a-SiC:H) films embedded with Ge nanoparticles (Ge NPs). The a-SiC:H films and Ge NPs are synthesized using different methods but in situ, in the same chamber, creating a layered structure. Structural analysis, including TEM, SAED, and EDX, is used to confirm the formation of Ge NPs, while optical properties are characterized using absorption spectroscopy and photoluminescence spectroscopy.

  1. The introduction is not clearly written; it is difficult to understand the specific objectives and motivations behind the research. It is unclear whether similar systems have been studied before and what knowledge gaps the authors are addressing. Is this a new synthesis method, a material with improved properties, or a novel system with measurable advantages?

The last paragraph of the Introduction was replaced by the following text to emphasize what knowledge gaps the authors are addressing:  

This study aims to develop a novel synthesis method for a-SiC:H films embedded with  Ge NPs and to investigate their structural and optical properties. The integration of Ge NPs into a-SiC:H films could potentially enhance their optical and electronic properties, making them suitable for advanced optoelectronic applications. While previous studies have explored the synthesis of a-SiC:H films and Ge NPs separately, there is limited research on the in-situ synthesis of these materials in a single chamber, which could offer better control over the material properties. This study introduces a novel in situ synthesis method that allows for the simultaneous formation of a-SiC:H films and Ge NPs, resulting in a unique layered structure with potentially superior properties.

  1. The composition of amorphous silicon the authors are interested in is not clear: what does x < 0,1 mean in the formula a-Si1-xCx:H? Does it imply 0.1? The authors then refer to the material as simply a-SiC:H, making comparison of the results with literature data problematic. What is the actual composition of the film used in the study?

The formula a-Si1-xCx:H has been removed from the Introduction and replaced with a-SiC:H consistent with the rest of the document.

  1. The sentence "It is challenging to measure low values of diffusion length in amorphous thin layers." appears abruptly at the end of the introduction and requires further elaboration.

The sentence "It is challenging to measure low values of diffusion length in amorphous thin layers” has been removed from the revised Introduction.

  1. Further, the rationale behind the choice of film thicknesses and the resulting optical properties is not clear. Why do the thinnest film (#1-Ge) and the thickest film (#2-Ge) contain dissolved Ge NPs, while sample #3 does not? What is the mechanism? Can the dissolution of Ge NPs be controlled by thickness, or what causes different sample outcomes? Additionally, the optical properties of Sample #1 are not shown later in the manuscript.

To improve clarity of the text the sample names #1, #2 and #3 (samples without Ge) were included in the Table 1. The word “sample #3-Ge” was added to the sentence “To prevent dissolution of Ge in a-SiC:H and create Ge NCs (sample #3-Ge)…” (line 82)

The word “sample #1” was removed from the sentence “The samples for TEM analysis of were deposited simultaneously under the same conditions as samples deposited on fused silica substrate, but directly onto standard (commercial) Cu TEM grids covered by carbon foil”  (line 87) followed by the sentence “The relatively thick sample #3 was deposited on fused silica substrate for optical measurements to evaluate the optical absorption edge.”

 The sentence “Samples #1, #2 and #3 were prepared under the same conditions without Ge evaporation.” Was moved from Table 1 caption to the main text and modified to “Samples #1, #2 and #3 were prepared during the same process as #1-Ge, #2-Ge and #3-Ge  but covered with shield during Ge evaporation.”

  1. The samples are not clearly identified in the manuscript. Table 1 presents only a few samples; it should be expanded to include all samples produced in this work for easier reference.

To improve clarity of the text the samples #1, #2 and #3 (samples without Ge) were included in the Table 1.

  1. Figure 2, depicting the sample structure, does not indicate that a carbon layer was deposited before Ge deposition. Was another C deposition performed afterward before the next layer of a-SiC:H? How does this C layer affect the overall composition of the film and subsequently its optical properties?

Figure 2 has been removed from the revised manuscript and the other figures have been renumbered.

  1. Page 3, line 90: Which samples are being referred to as "selected samples"?

The words “The selected samples” were replaced by “The samples”

  1. Figure 7 and the corresponding text: The authors use both "absorption" and "absorptance" when referring to the same effect. The terminology should be consistent.

The words “the optical absorption values” were replaced by “the optical absorption coefficient values” and “Ge increases NIR absorption” by “Ge increases NIR absorptance”

  1. Page 7: What is the purpose of annealing the film at 550 °C, and how does the conclusion about the effusion of H apply to the samples with Ge nanoparticles?

The high temperature annealing was our unsuccessful attempt to achieve post-deposition crystallization of Ge NCs. To make it clear for the reader we included into the Conclusions the following statement “We have shown that it is not possible to achieve post-deposition crystallization of Ge dissolved in a-SiC:H film by high-temperature annealing at 550 °C because a-SiC:H degrades at higher temperatures. Instead,  we have shown that the Ge NCs can be grown during deposition process at relatively low temperature at about 400 °C”

  1. Page 7, line 212: The authors should elaborate on how they determined the specific range of 10-30 nm for the free carrier diffusion length.

The specific range of 10-30 nm was replaced in the Discussion by words “few tens of nm. This estimation follows the observation that PL diminished due to the presence of Ge-induced non-radiative recombination in the very thin sample (thickness of a-SiC:H sub-layer 4 nm ) whereas it is partly preserved in thicker sample (thickness of a-SiC:H sub-layer 45 nm).”

Reviewer 2 Report

Comments and Suggestions for Authors

The paper presents a relevant study about the integration germanium nanocrystals 9 (Ge NCs) into hydrogenated amorphous silicon carbide (a-SiC:H) thin films.

In the "Materials and Methods" section, the description of the deposition technique should be improved so the reader can fully understand it. I would strongly recommend to draw an scheme of the system and include it before Figure 1. If the current photographs in Figure 1 are to be included, the different parts of the system should be identified in it, so they really give insights into the process.

The quality of both Figure 3  and Figure 6 should be improved.

Regarding the contents of Figure 6, it is not clearly stated which samples are analyzed in the results provided in the figure. Due to the bad quality of the image, the legends in the figures can not be read so it is hard to follow the description of the results in lines 168-176.

In section 3.3 (optical spectroscopy), the absorptance results of sample 1 are not found. The authors should justify this point or include measurements of such samples. A description of how the FilmWizard software was used should also be included.

I would also recommend the authors to organize better the results and discussion section. For example, after Figure 9, there is a discussion paragraph, but then a specific section for discussion is also included (Section 4).  In order to be consistent, I would suggest to follow one of the following two options:

1. Include discussions of each specific issue after describing the corresponding result.

2. Remove all the partial discussions made along the text and include all of them in the "discussion" section.

In section 4, the Tauc gap of sample 2 is used for discussing the PL values. Why has it not been measured for other samples and included in the discussion? In lines 215-216, it is stated that "the diffusion length typically depends  on the material’s quality". This statement should be replaced by a more specific, evidence-based one.

Author Response

REVIEWER II

The paper presents a relevant study about the integration germanium nanocrystals 9 (Ge NCs) into hydrogenated amorphous silicon carbide (a-SiC:H) thin films.

  1. In the "Materials and Methods" section, the description of the deposition technique should be improved so the reader can fully understand it. I would strongly recommend to draw an scheme of the system and include it before Figure 1. If the current photographs in Figure 1 are to be included, the different parts of the system should be identified in it, so they really give insights into the process.

The Figure 1 has been changed and  the different parts of the system were identified in it, so they really give insights into the process.

  1. The quality of both Figure 3 and Figure 6 should be improved.

The quality of both figures was improved and the resolution was upgraded to 300dpi .

  1. Regarding the contents of Figure 6, it is not clearly stated which samples are analyzed in the results provided in the figure. Due to the bad quality of the image, the legends in the figures can not be read so it is hard to follow the description of the results in lines 168-176.

The resolution of the Figure 6 (now Figure 4) was increased from 96 dpi to 300 dpi.

  1. In section 3.3 (optical spectroscopy), the absorptance results of sample 1 are not found. The authors should justify this point or include measurements of such samples. A description of how the FilmWizard software was used should also be included.

Only the optical absorption edge of a relatively thick sample #2 (thickness 90 nm ) could be evaluated from the optical absorption coefficient. The samples #1 and #3 were too thin and they were prepared mainly for TEM measurements. It makes no sense to evaluate the optical absorption coefficient of samples #1-Ge, #2-Ge and #3-Ge because these samples are composites of two different materials. The  description of how the FilmWizard software was used was included in the revised manuscript. “First the index of refraction was evaluated from the reflectance spectrum using parametric Lorenc oscillator model and then the optical absorption coefficient was evaluated at each energy independently from optical absorptance spectra measured by PDS”.

  1. I would also recommend the authors to organize better the results and discussion section. For example, after Figure 9, there is a discussion paragraph, but then a specific section for discussion is also included (Section 4). In order to be consistent, I would suggest to follow one of the following two options:
    Include discussions of each specific issue after describing the corresponding result.
    2. Remove all the partial discussions made along the text and include all of them in the "discussion" section
    .
    In section 4, the Tauc gap of sample 2 is used for discussing the PL values. Why has it not been measured for other samples and included in the discussion? In lines 215-216, it is stated that "the diffusion length typically depends  on the material’s quality". This statement should be replaced by a more specific, evidence-based one.

The results and discussion part was completely rewritten. The statement "the diffusion length typically depends  on the material’s quality" has been removed from the revised manuscript.

Reviewer 3 Report

Comments and Suggestions for Authors

The paper reported the synthesis and characterization of amorphous SiC layers with embedded Ge nanocrystals. The films exhibit interesting optical properties. I would recommend the publication of the paper after addressing the following issues.

1. Scale bars are needed in Figure 1 to demonstrate the sizes of the chamber and sample holder.

2. When describing sample #1, the EDX results were not given in the paper, which should be shown in the paper. Additionally, it would be better to provide the EDX mapping of the samples.

3. Figure 6b, 6c, and 6d should be redrawn, not using screenshots. They are too vague.

4. It is strange that no Ge nanocrystals were formed in sample #1 and #2. The authors should explain this phenomenon.

5. Why did the authors measured the absorbance spectra in different ranges for different samples? In addition, the absorbance spectra of sample #1 under various conditions were not shown, why?

6. Regarding the Tauc plot, why did the authors just give the plot of sample #2 without Ge layer?

Author Response

REVIEWER III

The paper reported the synthesis and characterization of amorphous SiC layers with embedded Ge nanocrystals. The films exhibit interesting optical properties. I would recommend the publication of the paper after addressing the following issues.

  1. Scale bars are needed in Figure 1 to demonstrate the sizes of the chamber and sample holder.

The scale bar was included in Figure 1.

  1. When describing sample #1, the EDX results were not given in the paper, which should be shown in the paper. Additionally, it would be better to provide the EDX mapping of the samples.

  1. Figure 6b, 6c, and 6d should be redrawn, not using screenshots. They are too vague.

The resolution of the Figure 6 (now Figure 4) was increased from 96 dpi to 300 dpi.

  1. It is strange that no Ge nanocrystals were formed in sample #1 and #2. The authors should explain this phenomenon.

The explanation was added into the revised manuscript into . Materials and Methods

2.1. PECVD, vacuum evaporation and thermal annealing:  “To prevent dissolution of Ge in a-SiC:H and create Ge NCs in the sample #2-Ge, the a-SiC:H surface was saturated with carbon prior the Ge evaporation using methane in hydrogen plasma (flow rate ration 100:2).”

  1. Why did the authors measured the absorbance spectra in different ranges for different samples? In addition, the absorbance spectra of sample #1 under various conditions were not shown, why?

Only the optical absorption edge of a relatively thick sample #2 (thickness 90 nm ) could be evaluated from the optical absorption coefficient. The samples #1 and #3 were too thin and they were prepared mainly for TEM measurements

  1. Regarding the Tauc plot, why did the authors just give the plot of sample #2 without Ge layer?

It makes no sense to evaluate the optical absorption coefficient of samples #1-Ge, #2-Ge and #3-Ge because these samples are composites of two different materials.

Round 2

Reviewer 1 Report

Comments and Suggestions for Authors

The authors significantly improved the presentation of their experiments and discussions. The paper is suitable for publication.

There are a few typos and inconsistencies to address:

  • In Figure 7, the label (a) should be removed. Also, please specify which samples are represented by the red, gray, and black lines in the PL spectra.
  • Line 90 contains a redundancy: "in situ deposited in situ by evaporation." Also, further text has some repetition. Please ensure the accompanying description is clear and concise.